# Comparative Analysis of the Symbiotic Microbiota in the Chinese Mitten Crab (*Eriocheir sinensis*): Microbial Structure, Co-Occurrence Patterns, and Predictive Functions

**DOI:** 10.3390/microorganisms11030544

**Published:** 2023-02-21

**Authors:** Jicheng Yang, Qianqian Zhang, Tanglin Zhang, Shuyi Wang, Jingwen Hao, Zhenbing Wu, Aihua Li

**Affiliations:** 1State Key Laboratory of Freshwater Ecology and Biotechnology, Institute of Hydrobiology, Chinese Academy of Sciences, Wuhan 430072, China; 2College of Fisheries and Life Science, Dalian Ocean University, Dalian 116023, China; 3National Aquatic Biological Resource Center (NABRC), Wuhan 430072, China; 4University of Chinese Academy of Sciences, Beijing 100049, China; 5School of Environmental Science and Engineering, Huazhong University of Science and Technology, Wuhan 430074, China

**Keywords:** Chinese mitten crab, symbiotic microbiota, hemolymph, hepatopancreas, intestine, high-throughput sequencing

## Abstract

Symbiotic microorganisms in the digestive and circulatory systems are found in various crustaceans, and their essential roles in crustacean health, nutrition, and disease have attracted considerable interest. Although the intestinal microbiota of the Chinese mitten crab (*Eriocheir sinensis*) has been extensively studied, information on the symbiotic microbiota at various sites of this aquatic economic species, particularly the hepatopancreas and hemolymph, is lacking. This study aimed to comprehensively characterize the hemolymph, hepatopancreas, and intestinal microbiota of Chinese mitten crab through the high-throughput sequencing of the 16S rRNA gene. Results showed no significant difference in microbial diversity between the hemolymph and hepatopancreas (Welch *t*-test; *p* > 0.05), but their microbial diversity was significantly higher than that in the intestine (*p* < 0.05). Distinct differences were found in the structure, composition, and predicted function of the symbiotic microbiota at these sites. At the phylum level, the hemolymph and hepatopancreas microbiota were dominated by Proteobacteria, Firmicutes, and Acidobacteriota, followed by Bacteroidota and Actinobacteriota, whereas the gut microbiota was mainly composed of Firmicutes, Proteobacteria, and Bacteroidota. At the genus level, *Candidatus Hepatoplasma*, *Shewanella*, and *Aeromonas* were dominant in the hepatopancreas; *Candidatus Bacilloplasma*, *Roseimarinus,* and *Vibrio* were dominant in the intestine; *Enterobacter*, norank_Vicinamibacterales, and *Pseudomonas* were relatively high-abundance genera in the hemolymph. The composition and abundance of symbiotic microbiota in the hemolymph and hepatopancreas were extremely similar (*p* > 0.05), and no significant difference in functional prediction was found (*p* > 0.05). Comparing the hemolymph in the intestine and hepatopancreas, the hemolymph had lower variation in bacterial composition among individuals, having a more uniform abundance of major bacterial taxa, a smaller coefficient of variation, and the highest proportion of shared genera. Network complexity varied greatly among the three sites. The hepatopancreas microbiota was the most complex, followed by the hemolymph microbiota, and the intestinal microbiota had the simplest network. This study revealed the taxonomic and functional characteristics of the hemolymph, hepatopancreas, and gut microbiota in Chinese mitten crab. The results expanded our understanding of the symbiotic microbiota in crustaceans, providing potential indicators for assessing the health status of Chinese mitten crab.

## 1. Introduction

Over the last couple of decades, the vital role of the microbiota in the survival, homeostasis, and development of animals has been elucidated [1]. The microbiota plays an important role in host health and metabolism, such as antagonizing pathogenic bacteria, regulating host immunity, and participating in nutrient absorption and vitamin synthesis [2]. At the same time, the composition structure and physiological function of microbiota are influenced by various external environments, food, the host’s own physiological state, and other factors [2,3].

Diverse symbiotic microbiota are important components of different parts of the host body, including the intestine, hemolymph, and hepatopancreas. The animal intestine is a complex and varied ecosystem containing a myriad of microorganisms that improve digestive efficiency and nutrient utilization, enhance the immune system, and inhibit the adhesion of opportunistic pathogens [4,5,6]. The hepatopancreas is a critical organ involved in the digestion and absorption of nutrients and plays an essential in innate immunity in invertebrates that also inhabits pathogens [7,8,9]. Hemolymph plays an important role in host immune process [10], which is generally sterile in vertebrates, but unique microbiota are present in the hemolymph of healthy crustaceans [11]. To date, little information about the microbiota of healthy crustaceans is available, let alone their comparative variations in different tissues.

The Chinese mitten crab (*Eriocheir sinensis)* is considered an invasive species in Europe and North America. In China, it is an economically valuable species and an important farming industry in many coastal cities and other areas. Owing to its delicious taste and rich nutrition, the Chinese mitten crab is widely loved by Chinese consumers [12]. As in many areas of animal nutrition, the microbiota’s role in the growth, health, and survival of cultured organisms have become a major subject of interest [13]. Therefore, research about symbiotic microbiota in healthy tissues will promote understanding of the role of microbiota in crab growth and health, and provide data support for targeted microbiota regulation to prevent diseases.

High-throughput sequencing technologies, in combination with bioinformatics approaches, are completely transforming microbiology, and they are broadly expanding our view of microbial life as it relates to invertebrates, such as microbial community composition, functional potential, and metabolic activity [14]. Therefore, in this study, the high-throughput sequencing technique was used in comparing the intestinal microbiota of the intestine, hemolymph, and hepatopancreas. The objectives of this study were as follows: (1) identification of core microbiota in the three sites of Chinese mitten crab, (2) identification of differences in symbiotic microbiota composition and characteristics of different parts of the Chinese mitten crab in healthy condition and their predicted functions, and (3) analysis of the inter-relationship of symbiotic microbiota among the three sites of the Chinese mitten crab.

## 2. Materials and Methods

### 2.1. Sample Collection and Preparation

Crabs were sourced from Beihe and Diaochahu aquaculture farms at Xijiang and Diaochahu County, Hanchuan City, Hubei Province, China. No disease symptoms were observed in the crab farms for 3 months before sampling. Twelve crabs of the nearly same size (crabs in Beihe; mean weight, 156.26 ± 7.23 g, crabs in Diaochahu; mean weight, 162.34 ± 8.28 g) were randomly sampled from two farms on 4 December 2021 (Appendix A). All samples were brought back to the laboratory as soon as possible in an ice incubator.

All crabs were anesthetized using an MS-222 (Sigma, Darmstadt, Germany) solution. The surfaces of the crabs were cleaned with sterile water to reduce bacterial contamination and then disinfected with 75% ethanol before dissection. Under sterile conditions, 1 mL of hemolymph was drawn with a 1.5 mL syringe (the samples from Beihe were marked as B1–B6, and the samples from Diaochahu were marked as B7–B12). Then, the hepatopancreas from the other tissues of the cephalothorax were carefully separated. The tissue from the surface of the hepatopancreas were put in a sterile centrifuge tube using sterile forceps. (The samples from Beihe were marked as H1–H6, and the samples from Diaochahu were marked as H7–H12). Then, the intestine was collected in sterile tubes (The samples from Beihe were marked as I1–I6, and the samples from Diaochahu were marked as I7–I12). All samples were snap-frozen in liquid nitrogen and kept frozen at −80 °C until used for genomic DNA extraction.

### 2.2. DNA Extraction

DNA extraction was executed according to the QIAamp^®^ DNA Blood & Tissue Kit (Qiagen, Germany) manufacturer’s instructions. (1) Pipet 20 μL proteinase K into a 2 mL sterile centrifuge tube. 100 μL hemolymph was added (25 mg to the hepatopancreas and intestine). The volume was adjusted to 220 μL with PBS (in the hepatopancreas and intestine, 180 μL Buffer ATL was added and mixed by vortexing, then incubated at 56 °C until completely lysed). (2) 200 μL Buffer AL was added, then mixed thoroughly by vortexing (incubate hemolymph samples at 56 °C for 10 min). (3) 200 μL ethanol (96–100%) were added and mixed thoroughly by vortexing. (4) The mixture was pipeted into a DNeasy Mini spin column placed in a 2 mL collection tube and centrifuged at 8000 rpm for 1 min. The flow-through and collection tube were discarded. (5) The spin column was placed in a new 2 mL collection tube, and 500 μL Buffer AW1 was added. The mixture was then centrifuged for 1 min at 8000 rpm. The flow-through and collection tubes were discarded. (6) The spin column was placed in a new 2 mL collection tube, and 500 μL Buffer AW2 was added, then the mixture was centrifuged for 3 min at 14,000 rpm. the flow-through and collection tubes were discarded. (7) the spin column was then transferred to a new 2 mL sterile centrifuge tube. (8) the mixture then was eluted DNA by adding 200 μL Buffer AE to the center of the spin column membrane and incubated for 1 min at room temperature. It was then centrifuged for 1 min at 8000 rpm. The concentration and purity of the extracted DNA were checked using a NanoDrop 2000 spectrophotometer (Thermo Fisher Scientific, Waltham, MA, USA) and stored in a −80 °C refrigerator for subsequent use.

### 2.3. 16S rRNA Gene High-Throughput Sequencing

The V3–V4 hypervariable regions of the bacterial 16S rRNA gene were amplified using primers 338F (5′-ACTCCTACGGGAGGCAGCA-3′) and 806R (5′-GGACTACHVGGGTWTCTAAT-3′). The PCR amplification conditions were as follows: denaturation at 98 °C for 30 s; 27 cycles at 98 °C for 15 s, annealing at 50 °C for 30 s, and elongation at 72 °C for 30 s; with a final extension at 72 °C for 5 min. PCR was performed in 25 μL of mixture containing 5 μL of 5× reaction buffer, 5 μL of 5× high GC Buffer, 2 µL of dNTPs (10 mM), 1 μL of each primer (10 μM), 0.25 μL of Q5 high-fidelity DNA polymerase, 2 µL of template DNA, and sterile water added to a volume of 25 μL. The resulting PCR products were extracted from 2% agarose gel and further purified using the AxyPrep DNA gel extraction kit (Axygen Biosciences, Union City, CA, USA). Purified amplicons were pooled in equimolar and paired-end sequenced simultaneously on the Illumina MiSeq PE300 platform (Illumina, San Diego, CA, USA) at the Wuhan Frasergen Bioinformatics Co., Ltd. (Wuhan, China).

### 2.4. High-Throughput Sequencing Data Analysis Processing

Paired-end raw sequencing sequences were quality-filtered using fastp (https://github.com/OpenGene/fastp, version 0.19.6, 5 September 2022) software and merged by FLASH (http://www.cbcb.umd.edu/software/flash, version 1.2.11, 5 September 2022) software. (1) The samples were demultiplexed according to the barcode and primers at the beginning and end of the sequence, and the sequence direction was adjusted. The number of mismatches allowed for barcode is 0, and the maximum number of primer mismatches is 2. (2) The 300 bp reads were truncated on any site with an average quality score of <20 over a 50 bp sliding window, and then the truncated reads that were <50 bp were discarded; reads containing N-bases were also removed. (3) According to the overlapping relationship between PE reads, the pairs of reads were merged into a sequence with a minimum overlap length of 10 bp. (4) The maximum mismatch ratio allowed in the overlap area of the merged sequence was 0.2.

### 2.5. Bioinformatics and Statistical Analysis

All samples at the 3 sites were randomly resampled to 47,419 reads based on the minimum number of sequences. Operational taxonomic units (OTUs) with 97% similarity [15] cutoff [16] were clustered using UPARSE (http://drive5.com/uparse/, 9 September 2022) version 7.0 [16], and chimeric sequences were identified and removed. The taxonomy of each OTU representative sequence was analyzed by RDP Classifier (http://rdp.cme.msu.edu/, 9 September 2022) version 2.2 [17] against the Bacterial Silva 16S rRNA database (SILVA SSU 138). Alpha diversity was estimated using the richness indices Chao and Sobs, and diversity indices Simpson and Shonnon, and Good’s coverage (coverage). Principal coordinates analysis (PCoA), hierarchical clustering tree and analysis of similarity based on Bray-Curtis metric (ANOSIM) were used to analyze Beta-diversity. The linear discriminant analysis (LDA) effect size (LEfSe) was used to distinguish differences in genomic characteristics under different biological conditions [18]. Microbial biomarkers and functional differences were identified through LEfSe (http://huttenhower.sph.harvard.edu/galaxy/; *p* < 0.05 and LDA score >3.0, 12 September 2022) analysis [19]. Tax4Fun was used for predicting the function of bacterial communities. Welch’s *t*-test was used to analyze differences between the two independent groups, and the Kruskal-Wallis H-test was used to assess differences between multiple independent groups (SPSS, version 22.0) [20]. The *p*-value was corrected following the Bonferroni method.

We used the Hmisc package for calculating correlation networks in R, and corrections for multiple comparisons were executed for Spearmans’s rank correlations using the false discovery rate correction [21]. Correlation coefficients >0.7 and corresponding *p*-values < 0.05 were regarded as statistically stable and were admitted to generate the network. The network structure was explored visually using Gephi, using undirected networks and Fruchterman-Reingold format for the layout. The relevant topological parameters and node scores of the network [22] were obtained in Gephi.

## 3. Results

### 3.1. Diversity and Structure of Bacterial Communities in Different Sites

After quality filtration, a total of 2,479,691 effective sequences were obtained from 36 samples with an average read length of 457 bp. Good’s coverage showed that more than 98% of the bacterial species in the samples were detected (Appendix A), and the rarefaction curves tended to approach the saturation plateau, indicating that the majority of the microbial diversity present in the samples was detected (Appendix A). The microbial complexity of the three sites was assessed using alpha diversity indices of the taxonomic profiles at the 97% similarity threshold (Figure 1). Statistical analysis that showed no significant differences (Welch *t*-test, *p* > 0.05) were found in the diversity and richness of the bacterial communities between the hemolymph and hepatopancreas, but they were significantly higher than those in the intestine (Welch *t*-test, *p* < 0.01).

Multivariate statistical analyses were performed to compare the integral structure of the bacterial communities in different sites. ANOSIM revealed significant differences (R = 0.67, *p* = 0.001) in the bacterial community structures among the three sites (Appendix A). The PCoA plot visualized the ANOSIM results, which showed intestine samples were tightly aggregated and separated from hemolymph and hepatopancreas samples. Hemolymph samples were clustered into two parts, and hepatopancreas samples were dispersed. Overall, the two principal coordinates obtained from the PCoA explained 33.58% of the variations among all samples (Figure 2A). The hierarchical clustering tree on the OTU level disclosed that the bacterial communities in the intestines clustered into one branch, hepatopancreas and hemolymph samples were clustered into many small branches separately, and some samples were not completely separated (Figure 2B).

### 3.2. Taxonomic Composition of the Bacterial Communities in Different Sites

The taxonomic classification of sequences from all samples was assigned to 66 bacterial phyla, 199 classes, 505 orders, 902 families, 2111 genera, and 19,521 OTUs (Appendix A). At the phylum level, Proteobacteria (34.74% ± 5.26%) was the most abundant phylum in hemolymph and followed by Firmicutes (15.10% ± 2.06%), Acidobacteriota (10.63% ± 2.24%), Bacteroidota (9.35% ± 2.87%), Actinobacteriota (6.06% ± 1.66%), and the relative abundances of the remaining phyla were less than 5% (Figure 3; Appendix A). The dominant phyla in hepatopancreas were Proteobacteria (34.71% ± 18.42%) and followed by Firmicutes (25.73% ± 16.10%), Acidobacteriota (9.23% ± 3.37%), Bacteroidota (8.18% ± 3.84%), and the relative abundances of the remaining phyla were less than 5% (Figure 3; Appendix A). The dominant phyla in the intestine were Firmicutes (41.37% ± 23.49%) and followed by Proteobacteria (29.50% ± 18.42%), Bacteroidota (23.06% ± 10.61%), and Acidobacteriota (1.98% ± 0.96%), and the relative abundance of the remaining phyla was less than 1% (Figure 3; Appendix A).

At the genus level, as shown in the heatmap (Appendix A), the overall genus relative abundance in the hemolymph was low (relative abundance <5%). The relative abundance of unclassified_norank_Bacteria, *Enterobacter,* norank_Vicinamibacterales, and *Pseudomonas* was relatively high. The dominant genus in hepatopancreas was *Candidatus Hepatoplasma* and followed by *Shewanella*, *Aeromonas*, *Enterobacter*, and *Enterococcus*, and the relative abundance of the remaining genera was low. *Candidatus Bacilloplasma* was the dominant phylotype in the intestine and followed by unclassified_Alphaproteobacteria, *Roseimarinus*, *Vibrio*, *Dysgonomonas*, and *Acinetobacter*.

### 3.3. Differences of the Bacterial Communities at Different Sites

At the genus level, the microbiota at the three sites of 12 crabs was analyzed by a petalogram. The results showed that there were 226 genera shared by hemolymph and hepatopancreas, 187 genera shared by hemolymph and intestine, and 204 genera shared by hepatopancreas and intestine, and 177 genera shared by the three sites (Appendix A). Petalogram analysis of each site of the 12 crabs showed 616 genera in hemolymph, of which 279 were shared genera, accounting for 17.26%; 1919 genera in hepatopancreas, of which 305 were shared genera, accounting for 15.83%; and 1443 genera in the intestine, of which 219 were shared genera, accounting for 15.18% (Figure 4A).

We further confirmed the presence of different OTUs in different sites by LEfSe. LEfSe identified 24 discriminative features (LDA score >3, Figure 5A) among three sites, in which nine OTUs were significantly enriched in the intestine, eight OTUs were significantly enriched in hepatopancreas, seven OTUs were significantly enriched in the hemolymph. *Shewanella*_OTU13154, *Pedobacter*_OTU17557, and *Nitrospira*_OTU7539 were significantly enriched in the hepatopancreas. The significantly enriched OTUs in the intestine were all from Proteobacteria, Firmicutes, and Bacteroidota, and *Candidatus Bacilloplasma*_OTU10971 from Firmicutes was the most abundant. The OTUs significantly enriched in the hemolymph group were all from different phyla, and *Chujaibacter*_OTU1225 and *Acinetobacter*_OTU11523 were from Proteobacteria.

### 3.4. Analysis of Bacterial Co-Occurrence Networks at Different Sites

Network complexity varied considerably across the three sites, compared with the intestine and hepatopancreas, microbial communities in the hemolymph had a more complex network (Figure 4B) with more edges (36,325), a higher average degree (36.108), and a high average clustering coefficient (0.819). The intestine had a higher mean clustering coefficient (0.707) compared with the hepatopancreas, although there were few nodes and edges (Appendix A). We used the degree as the basis for whether it was a keystone species, and the results showed (Appendix A) that the majority of keystone taxa belonged to Proteobacteria and Actinobacteria. Further analysis revealed that the keystone taxa at the genus level in the network were not the genus with high relative abundance in each site and most of them can be found in the shared genera. To evaluate the impact of the removal of keystone OTUs on networks, we constructed site-specific networks without them. The results showed that microbial network complexity was diminished without the keystone OTUs (Appendix A). For example, after removing the top 3% of potentially critical OTUs, important parameters such as the number of nodes, degree, and modularity, decreased (hemolymph, 30 OTUs; hepatopancreas, 23 OTUs; and intestine, 5 OTUs).

### 3.5. Functional Prediction of the Bacterial Communities

ANOSIM revealed that no significant difference (Appendix A, *p* > 0.05) was found between hemolymph and hepatopancreas, but a significant difference was found in the intestine (*p* < 0.05, Appendix A). Further microbial functions of the bacterial microbiota were predicted by Tax4Fun, and at the KEEG level 2 (Figure 6), hemolymph and hepatopancreas were enriched for cell growth and death, circulatory system, cardiovascular disease, infectious disease viral; the microbial functions enriched in the intestine were glycan biosynthesis and metabolism, signaling molecules and interaction, immune system, transport and catabolism, and cellular community eukaryotes. The functional clustering analysis based on average neighbor (unweighted pair-group method with arithmetic means) showed that the hepatopancreas and hemolymph functions were clustered into one branch, whereas the intestine was clustered at the other (Figure 6).

At KEGG level 3 (Figure 5B), we found metabolism, amino acid metabolism, xenobiotic biodegradation and metabolism, energy metabolism, metabolism of terpenoids and polyketides, and oxidative phosphorylation were significantly enriched in the hemolymph. The only microbial functions significantly enriched in hepatopancreas were porphyrin and chlorophyll metabolism. Environmental information processing, glycan biosynthesis and metabolism, human disease, and bacterial infectious disease were found significantly enriched in the intestine.

## 4. Discussion

Microorganisms are commonly found on the surface or in the body cavities of animals, such as humans, terrestrial mammals, fish, and other higher vertebrates [23,24,25]. By contrast, little is known about the microbiota of some invertebrates (such as crabs, shrimps, and molluscs) [26]. Studies on the symbiotic microorganisms of the Chinese mitten crab have mainly focused on the intestine [27,28,29], and little information is known about the microorganisms in the hemolymph and hepatopancreas [30]. Therefore, this study was the first to analyze the symbiotic microbiota of three sites in the intestine, hemolymph, and hepatopancreas of the Chinese mitten crab to improve the understanding of the symbiotic microbiota of crab.

To date, distributions of microbial communities at different sites have been reported in a variety of crustaceans, including aquatic crustaceans [31], *Daphnia magna* [32], shrimp, and crabs [33]. The results of diversity analysis in this study (Figure 1; Figure 2) showed that the hepatopancreas and hemolymph microbiota were more similar in structure and differed significantly from the intestine. This significant difference in microbiota structures may be due to the different functions of microorganisms at different sites [34]. The most important function of the intestine was the digestion and absorption of nutrients [27], and owing to the direct contact with the external environment, the microbiota was influenced by the environment, and the structure of the microbiota differs significantly from the hepatopancreas and hemolymph. Although hemolymph and hepatopancreas perform different functions, some studies have confirmed that the digestive system (including the midgut and hepatopancreas) is a potential source of hemolymph microorganisms [35], which may explain the similar microbiota structures of hemolymph and hepatopancreas.

In this study (Figure 3), we found that the dominant phylum in the intestine of Chinese mitten crab was Firmicutes, followed by Proteobacteria, while the relative abundance of Proteobacteria was the highest in the hepatopancreas and hemolymph, followed by Firmicutes, consistent with previous studies [27,28,30]. Such significant differences in microbial composition can be attributed to functional heterogeneity. Firmicutes include many probiotics that participate in the digestive process by the release of a variety of digestive enzymes, preventing intestine disease by inhibiting the adhesion of pathogens to the aquatic animal, modulating the immunity function of the intestinal mucosa, and maintaining intestinal barrier function [36]. Proteobacteria contain a large number of opportunistic pathogens, and their existence may contribute to the stimulation of the development of the host immune system and the maintenance of normal immune function [37]. The relative abundance of Bacteroidota and Actinobacteriota was second to Firmicutes and Proteobacteria among the three sites. Bacteroidetes have been reported to participate in carbohydrate transport and protein metabolism, which were significant for digesting diet [38], and Actinobacteria can produce various potent antibiotics that can inhibit the growth of intestinal pathogenic bacteria [39]. Thus, Bacteroidota was more abundant in the intestine, and Actinobacteriota was more abundant in the hemolymph and hepatopancreas.

This study revealed the differences between the three sites at the genus level from species composition and difference studies. *Candidatus Bacilloplasma* was detected as the dominant genus in the intestine (Appendix A), consistent with previous studies [28,40], and they suggested that *Candidatus Bacilloplasma* was a member of the “indigenous” population of the intestine [40]. *Candidatus Bacilloplasma* is a symbiont involved in the digestive system of crustaceans and is considered a new lineage of terrestrial isopod *Porcellio scaber* [41,42]. Previous studies have reported that *Candidatus Bacilloplasma* is related to the digestive function of the Chinese mitten crab [43], whether *Candidatus Bacilloplasma* is a potential probiotic, and its status and function in the intestinal bacteria remain to be further explored. *Candidatus Hepatoplasma* was the dominant genus in the hepatopancreas and had no significant difference from the intestine. Studies have shown that terrestrial isopods with *Candidatus Hepatoplasma* in their intestine have a higher survival rate when food and nutrition were scarce [44]. Therefore, *Candidatus Hepatoplasma* is likely to be a potential probiotic for crustaceans, and its function needs to be further explored. Significant difference in *Shewanella* was found among the three sites (Appendix A). Previous studies have confirmed the antimicrobial activity of certain *Shewanella* against bacterial pathogens, such as *Vibrio haemolyticus* and *Vibrio parahaemolyticus* [45], these findings suggested that *Shewanella* may represent important members of the hepatopancreas and contribute to defense against pathogens. In addition, the relative abundance of hemolymph microbiota was low in this study (Appendix A), and many of them were opportunistic pathogens, such as *Pseudomonas*, *Enterobacter*, *Acinetobacter*, and *Vibrio*. This microbiota in the hemolymph may function as a mock but constant stimulation to the immune system. The low albeit and opportunistic bacteria slightly stimulate immune molecules and cells to generate basic immunity in the host. Prompting the immune system to produce some products, such as some AMPs, lectins, and other antimicrobial components that both restrict symbiotic bacteria and resist external pathogens [11].

Core microbiota were long-term stable colonies in the host intestine and was important for physiological metabolism, nutrient absorption, and immune response [40]. In the present study, 12 randomly selected crabs of different sexes from two different regions, the results of each Petalogram (Figure 4A) showed 279 shared genera in the hemolymph, 305 shared genera in the hepatopancreas, and 219 shared genera in the intestine, which were likely to be the core microbiota of the Chinese mitten crab. Interestingly, except for *Shewanella* in the hepatopancreas, the top five genera in relative abundance in each site were in the shared genus. *Aeromonas*, *Candidatus Bacilloplasma*, and *Dysgonomonas* have been identified as core microbiota in the intestine in previous studies [40,42]. Core microbiota is considered the only host-related and not related to different growth stages, food, and environment, and their high percentage in the host means that the microbiota in the host is less affected by external influences [40]. In this study, the hemolymph had the highest percentage, followed by the hepatopancreas and the intestine had the least. This result suggested that in response to changes in the external environment, hemolymph stability was the highest, and the intestine was the lowest.

Network analysis can be used in studying microbial interactions and statistically identifying the taxa that are highly connected in the community [46,47,48]. A better understanding of the mechanisms of microbial action in the organism and insight into the interconnections between bacteria are needed. Our study showed that the complexity of the network was highest in hemolymph (Figure 4B; Appendix A), which may be because microorganisms were tightly regulated and controlled by the host’s basal immunity, in which the mechanisms may include hemocytes phagocytosis, agglutination, and production of reactive oxygen species and antimicrobials [11,49], contributing to the formation of a complex network of relationships among the microbiota. In addition, enhanced allelopathic interactions among microorganisms can promote the formation of a barrier against pathogen invasion [50]. Thus, under the pressure of host regulation and external pathogens, the hemolymph microbiota forms a complex network and structure. Interestingly, the intestine, which is a part of the digestive system, has a much simpler structure than the hepatopancreas microbiota network (Figure 4B; Appendix A). The possible reason is that the intestine is in direct contact with the external environment, and different physiological states, living environments, and dietary habits may lead to the ecological niche of the originally dominant microbiota being occupied [51,52,53,54], and changes in microbiota abundance and interactions. This changing external pressure results in a simpler network structure of the intestinal microbiota compared with the hepatopancreas. In this study, we first statistically identified the potential keystone taxa according to their connections and central positions in the networks, and the keystone taxa obtained from the network analysis were not of high relative abundance. This result is consistent with the fact that dominant taxa influence microbiome functioning owing to their high relative abundance, whereas keystone taxa may be able to selectively alter specific members of the microbiome and may thus exert their influence irrespective of their abundance [55]. Most of these keystone taxa were found in shared genera, which reminded us that shared genera may play an important role in organisms and require further study. Network analysis may not always represent true biotic interactions [56]. Therefore, the identification of keystone taxa through network analysis is “statistical” and requires keystone taxa empirical validation [57].

The results of intestinal microbial function prediction showed that the microorganisms of the intestine were significantly enriched in environmental information processing, glycan biosynthesis and metabolism, human disease, and bacterial infectious disease (Figure 5B) owing to the direct exposure of the intestine to the complex external environment, where various pathogenic bacteria can reach the intestine directly, and the intestinal microbiota assists in processing environmental information and fighting against external pathogenic bacteria. In addition, glycan biosynthesis and metabolism were significantly enriched in the intestine, suggesting that the intestinal microbiota can assist in the digestion and absorption of nutrients in the intestine. Microorganisms in the hepatopancreas were significantly enriched in porphyrin and chlorophyll metabolism (Figure 5B), and porphyrin can bind to metal ions and eliminate enriched metal ions in the hepatopancreas during metabolism [58]. Microorganisms in the hemolymph were significantly enriched in the metabolism of terpenoids and polyketides, xenobiotic biodegradation, and metabolism (Figure 5B). Terpenoids and polyketides can inhibit bacteria and can assist in maintaining stable hemolymph microbiota [59], and xenobiotic biodegradation and metabolism can eliminate external pathogenic bacteria [34]. This suggested that the hemolymph microbiota can degrade exogenous contaminants or toxic substances, thus preventing the hemolymph from exogenous damage. The amino acid metabolism, energy metabolism, and oxidative phosphorylation were significantly enriched in the hemolymph, suggesting that hemolymph microbiota can provide the substances and energy required for the immune process. This study reveals the differences in microbial functions at different sites of Chinese mitten crab, thereby further elucidating the unique function of symbiotic microbiota at different sites. However, the Tax4Fun remains limited in identifying consistent and differentially abundant functions. Therefore, further metagenomic studies should be performed to identify the unique function of symbiotic microbiota at different sites.

## 5. Conclusions

This study is the first to comprehensively characterize the taxonomic and functional differences in the hemolymph, hepatopancreas, and gut microbiota in Chinese mitten crab. The microbial structure, composition, and predicted function of the symbiotic microbiota were significantly different across these sites. Interestingly, the hemolymph had less variation in bacterial composition among individuals compared with the intestine and hepatopancreas. The health-specific bacteria might provide potential bacterial indicators for monitoring the health status of crabs. The microbial co-occurrence patterns varied greatly across the three sites, among which the hepatopancreas was the most complex, followed by hemolymph, and the gut had the simplest network. Furthermore, the functional pathways related to xenobiotic biodegradation and metabolism, energy metabolism, and metabolism of terpenoids and polyketides were significantly enriched in the hemolymph, whereas the functional pathways related to environmental information processing, glycan biosynthesis and metabolism, human disease, and bacterial infectious disease were significantly enriched in the hepatopancreas, and porphyrin and chlorophyll metabolism were significantly enriched in the intestine. These findings provide new clues for exploring the potential role of the symbiotic microbiota in maintaining crab health and affecting disease.

## Figures and Tables

**Figure 1 microorganisms-11-00544-f001:**
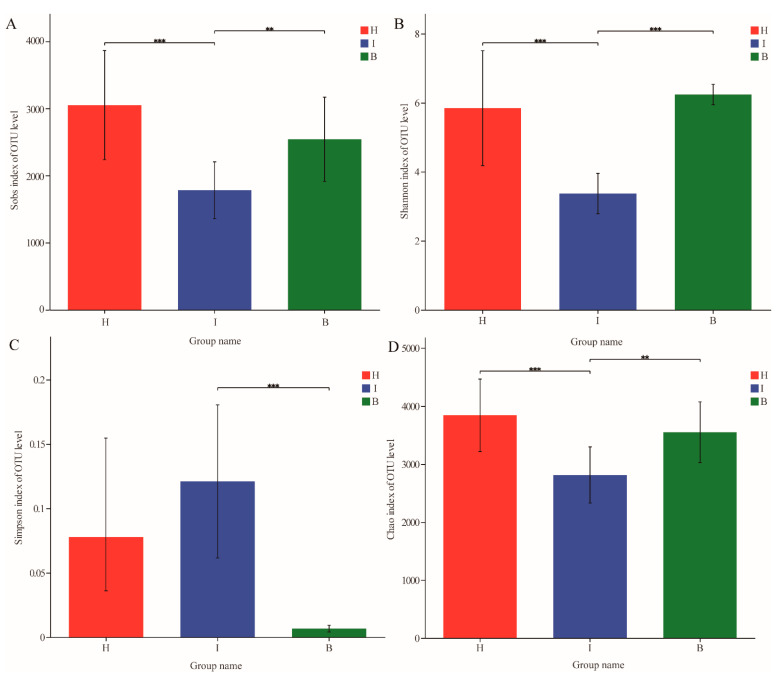
Comparison of alpha diversity indices of the bacterial communities at different sites. (**A**) Comparison of the Sobs index of the bacterial communities at different sites, (**B**) comparison of the Shannon index of the bacterial communities at different sites, (**C**) comparison of the Simpson index of the bacterial communities at different sites, and (**D**) comparison of the Chao index of the bacterial communities at different sites. Higher Sobs and Chao values indicate a higher richness; higher Shannon and lower Simpson values indicate a higher diversity. Statistical significances between the two sites were considered at ** *p* < 0.01, and *** *p* < 0.001 by Welch *t*-test. B: hemolymph (B1–B12); H: hepatopancreas (H1–H12); I: intestine (I1–I12).

**Figure 2 microorganisms-11-00544-f002:**
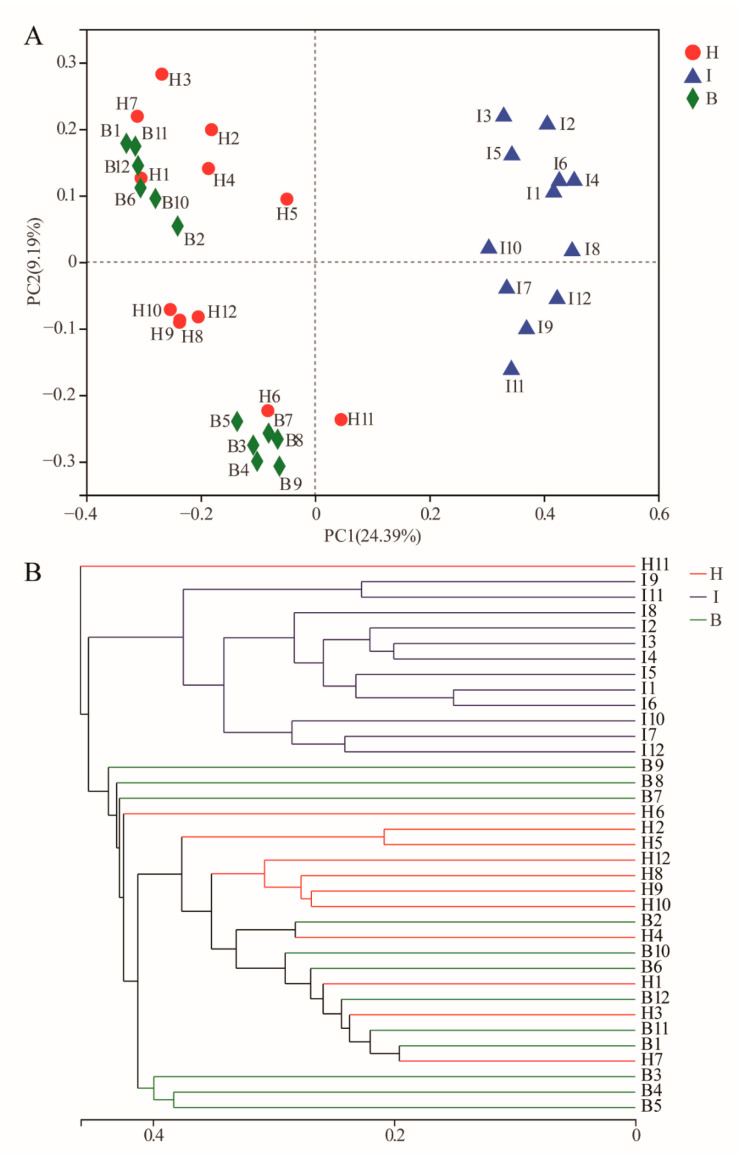
Analysis of the bacterial community structure at different sites. (**A**) Principal coordinate analysis based on the Bray-Curtis metric of the bacterial communities. The percentages indicate the relative contribution of the principal components. (**B**) The taxonomic clustering tree based on Bray-Curtis metric of the bacterial communities. B: hemolymph (B1–B12); H: hepatopancreas (H1–H12); I: intestine (I1–I12).

**Figure 3 microorganisms-11-00544-f003:**
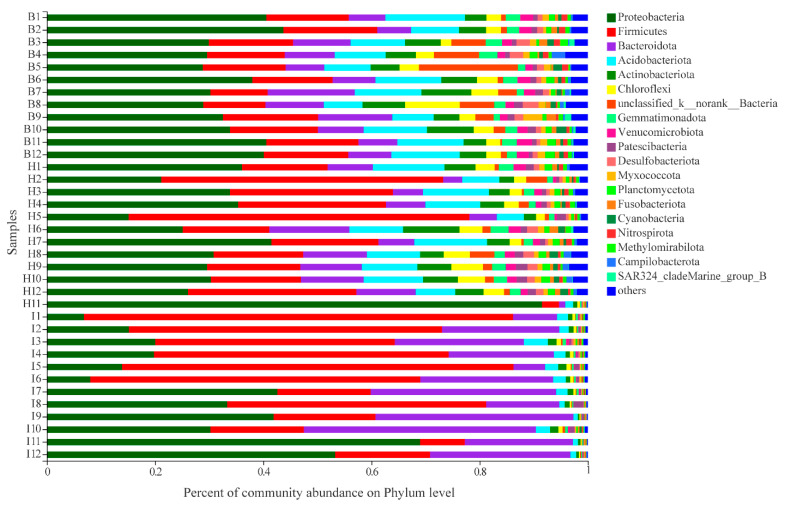
Distribution of the bacterial communities in all thirty-six samples at the phylum level. B: hemolymph (B1–B12); H: hepatopancreas (H1–H12); I: intestine (I1–I12).

**Figure 4 microorganisms-11-00544-f004:**
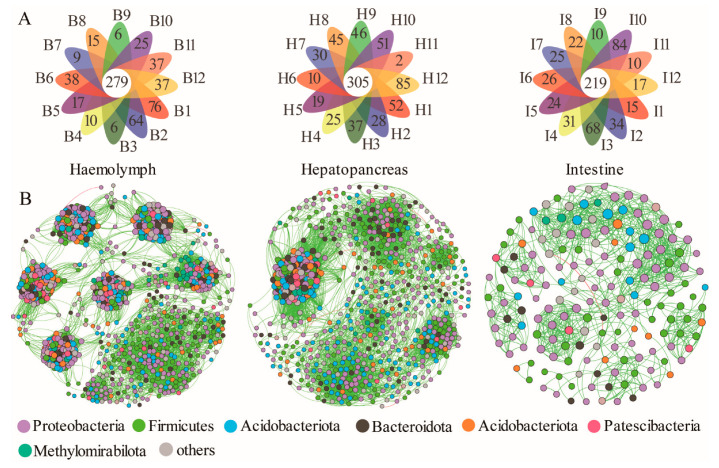
(**A**) The petalogram shows the number of genera that were shared at different sites of twelve crabs. (**B**) Network of co-occurring bacterial OTUs across three sites. Only Spearman’s correlation coefficient (r > 0.7 or r < −0.7 significant at *p* < 0.05) is shown. The nodes are colored according to phyla. Green edges represent positive correlations and red edges represent negative correlations. Node size is proportional to the betweenness centrality of each OTU, and edge thickness is proportional to the weight of each correlation.

**Figure 5 microorganisms-11-00544-f005:**
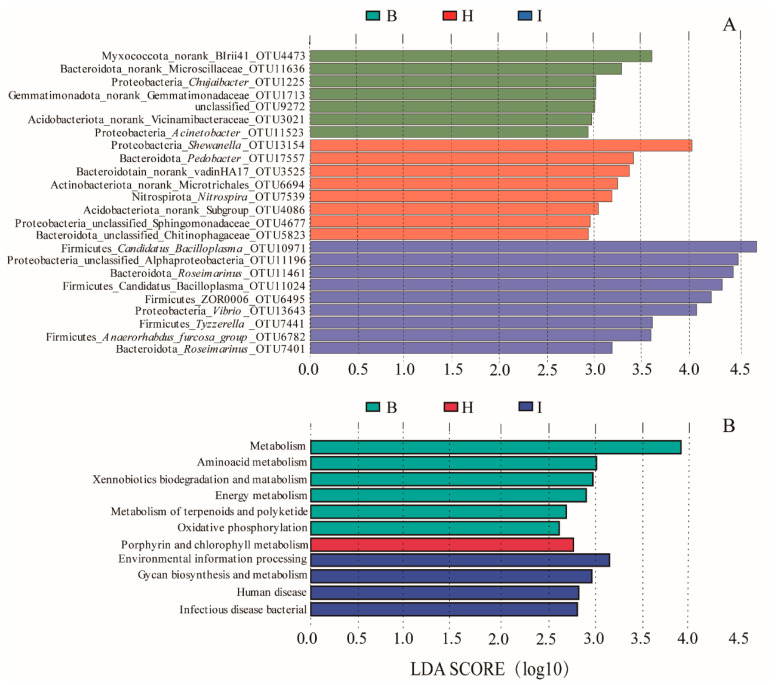
(**A**) LEfSe showing differences in the bacterial communities at the OTU level for three sites. (**B**) LEfSe showing differences in the predicted functions of different sites as predicted by Tax4Fun analysis. B: hemolymph (B1–B12); H: hepatopancreas (H1–H12); I: intestine (I1–I12).

**Figure 6 microorganisms-11-00544-f006:**
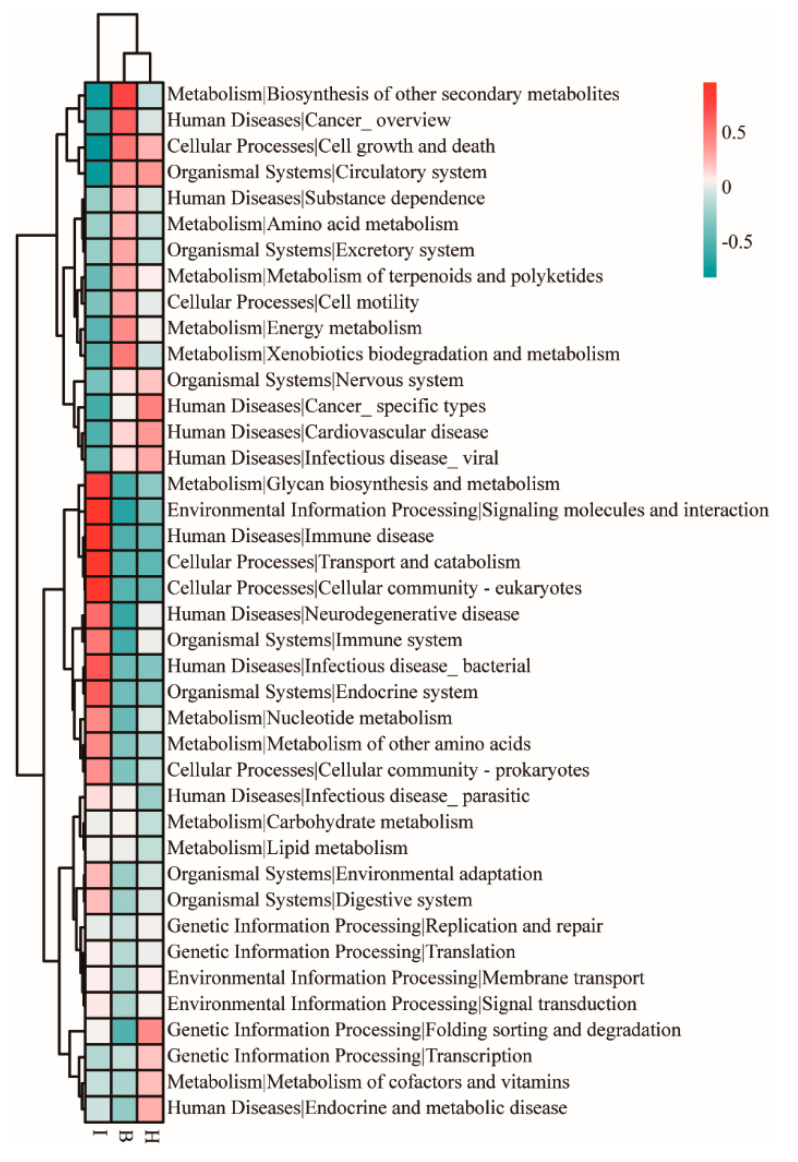
Heatmap profiles showing the functional categories (Kyoto Encyclopedia of Genes and Genomes (KEGG) level 2) of the bacterial communities, as predicted by Tax4Fun analysis. The functional clustering analysis is based on the unweighted pair-group method with arithmetic means (UPGMA). Rows represent the KEGG Orthology (KO) functions, columns represent the 36 samples, and the color intensity in the heatmap represents the relative abundance of the functional categories. B: hemolymph (B1–B12); H: hepatopancreas (H1–H12); I: intestine (I1–I12).

## Data Availability

The raw sequencing data was deposited in the NCBI Sequence ReadArchive under accession number PRJNA916675 (http://www.ncbi.nlm.nih.gov/sra, 29 December 2022).

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
