# Peer review of "Comparative Analysis of the Symbiotic Microbiota in the Chinese Mitten Crab (Eriocheir sinensis): Microbial Structure, Co-Occurrence Patterns, and Predictive Functions"

_microorganisms, 2023, doi:10.3390/microorganisms11030544_

Round 1
Reviewer 1 Report
Authors characterized the microbiota in 3 different locations (hepatopancrease, hemolymph, and intestine) of Chinese mitten crab 16S rRNA metabarcoding process. Overall they followed the straightforward method pipeline and performed intensive analyses. They found the significant difference in intestine compared to other two sites in this study. However, although all index indicated the enough coverage of sequence data, it still seemed that lower microbiota diversity in the intestine could be caused by lower sequence reads based on your rarefaction curve. Is there any reference data of intestine microbiota in other crustacean that can compare with your data. And data set in this study have information of locations that crab were collected. It would be interesting if there is any difference between locations too.
There are some minor comments here.
· Abstract is too long and need to be concise. L34-44 is not so necessary contents for Abstract.
· L79-82: Please check the grammar.
· L97-98: Delete “ All crabs appeared to be healthy and”. This made the author didn’t have confidence to your crab health. Saying that Crabs didn’t have symptoms is enough.
· L104: change crayfish to crabs
· L122: change spaces between numbers and title
· L129: change ‘polymer’ to ‘polymerase’
· L137- 147: This whole sequencing data process needs to be re-written. Way it was written are logically wrong and order of data preprocessing are messed up. Please ask someone who know the process and re-write.
· L147-151: this part better move to ‘Bioinformatics and statistical analysis’ section.
· L191: please rephrase “ but were significantly higher than intestine “ because grammatically nonsense.
· Fig2. A and B showed basically same information so that move Fig2 b to suppl data. And could you also add the crab collection information on the PCA graph?
· L220: Change ‘phylogenetic’ to ‘taxanomic’.
· L317-331: Please delete this paragraph. This is better fit to introduction.
· L345-346: Please delete.
Author Response
Please see the attachment。

Reviewer 2 Report
This article characterized the hemolymph, hepatopancreas, and intestinal microbiota of Chinese mitten crab through the high-throughput sequencing of the 16S rRNA gene. The results revealed the taxonomic and functional characteristics of the hemolymph, hepatopancreas, and gut microbiota in Chinese mitten crab. This manuscript is interesting, however it requires some revision before being acceptable for publication.
1. In the introduction section, the Chinese mitten crab Latin name (Eriocheir sinensis) appeared two times. It just be shown in the first time.
2. In the sample collection and preparation, “All crabs appeared to be healthy, and no disease symptoms were observed in the crab farms for 3 months before and after sampling ”. The samples were collected on December o4. As usual, most of the crab was captured and sold on Dec. In addition, the healthy crab was not susceptible to infection in winter. The sentence was suggested to modified as “……months before sampling”
3. In the DNA extraction section, how to extract the hemolymph microorganism DNA. As usual, the DNA is extracted from the precipitate by centrifugation according to the manufacturer’s instructions, however, the supernatant should also include the microorganism. Simply describe the procedure.
4. In the discussion section, “The composition of intestinal microbiota was greatly affected after hepatopancreas necro-sis, especially that the relative abundance of Candidatus Bacilloplasma was significantly reduced. These findings suggested that Shewanella may represent important members of the shrimp digestive tract and contribute to defense against pathogens.” These sentences seems irrelevance in context.
5. The female and male mature crab hepatopancreas is totally different in content and nutrition. The hepatopancreas microorganism between female and male crab was suggested to analyze.
6. The resolution of the pictures should be improved.
Round 2
Reviewer 1 Report
Authors have addressed well in the minor comments that I made. However, I couldn't find the response to main issue related to low sequencing reads in intestine and comparable microbiota data in my previous review. Please check the previous comments and address that.
And another thing, for method section 2.4, please change "distinguished" to "demultiplexed". and move sentences after (4) (Line 147-50) to the beginning of the whole section. This is the very beginning process right after sequencing.
